# Realization of Amyloid-like Aggregation as a Common Cause for Pathogenesis in Diseases

**DOI:** 10.3390/life13071523

**Published:** 2023-07-07

**Authors:** Soumick Naskar, Nidhi Gour

**Affiliations:** Department of Chemistry, Indrashil University, Kadi, Mehsana 382740, Gujarat, India

**Keywords:** amyloid, aggregation, self-assembly, gene mutation, disease pathogenesis, inborn errors of metabolism

## Abstract

Amyloids were conventionally referred to as extracellular and intracellular accumulation of Aβ42 peptide, which causes the formation of plaques and neurofibrillary tangles inside the brain leading to the pathogenesis in Alzheimer’s disease. Subsequently, amyloid-like deposition was found in the etiology of prion diseases, Parkinson’s disease, type II diabetes, and cancer, which was attributed to the aggregation of prion protein, α-Synuclein, islet amyloid polypeptide protein, and p53 protein, respectively. Hence, traditionally amyloids were considered aggregates formed exclusively by proteins or peptides. However, since the last decade, it has been discovered that other metabolites, like single amino acids, nucleobases, lipids, glucose derivatives, etc., have a propensity to form amyloid-like toxic assemblies. Several studies suggest direct implications of these metabolite assemblies in the patho-physiology of various inborn errors of metabolisms like phenylketonuria, tyrosinemia, cystinuria, and Gaucher’s disease, to name a few. In this review, we present a comprehensive literature overview that suggests amyloid-like structure formation as a common phenomenon for disease progression and pathogenesis in multiple syndromes. The review is devoted to providing readers with a broad knowledge of the structure, mode of formation, propagation, and transmission of different extracellular amyloids and their implications in the pathogenesis of diseases. We strongly believe a review on this topic is urgently required to create awareness about the understanding of the fundamental molecular mechanism behind the origin of diseases from an amyloid perspective and possibly look for a common therapeutic strategy for the treatment of these maladies by designing generic amyloid inhibitors.

## 1. Introduction

In 1838, German botanist Matthias Schleiden, for the very first time, introduced the term “amyloid” to the scientific literature to represent the amylaceous constituent of plants [1]. Later in 1854, Rudolph Virchow coined the term “amyloid” to describe the tissue abnormality that displayed an iodine-staining reaction [2]. Further, in 1907, German psychiatrist Aloysius (Alois) Alzheimer referred to plaques and neurofibrillary tangles present in the brain of patients with senile dementia as “amyloid” [3,4]. These abnormal aggregates were formed by the aggregation of Aβ42 and Aβ40 peptides, and the disease was later referred to as Alzheimer’s disease (AD) [5]. Subsequently, aggregation of tau oligomers was also considered a hallmark in the progression of AD [6]. Similar protein aggregation was also discovered in Parkinson’s disease (PD), wherein α-synuclein (α-Syn) protein aggregates inside the brain leading to pathogenesis like dementia and memory loss [7]. Further, it was also discovered that islet amyloid polypeptide (IAPP) aggregates to fibrillar structures leading to type II diabetes [8]. Research on studying the progression of the plethora of prion diseases suggests the formation of the infectious prion protein (PrP) with a tendency to cross-seed aggregation in other proteins [9,10]. Bovine spongiform encephalopathy (BSE or “mad cow” disease) in cattle, Creutzfeldt–Jakob disease (CJD) and variant CJD in humans, scrapie in sheep, and chronic wasting disease (CWD) in deer, elk, mice, and reindeer are some of the diseases associated with PrP aggregation [11]. A similar aggregation observed for the PrP protein was also noted for the p53 protein in the patho-physiology of cancer [12].

Several cross interaction studies of these amyloids have also been pursued [13,14] and functional materials have been designed using amyloid fibers due to their exceptional mechanical strength [15,16].

Initially, it was thought that amyloids are formed exclusively by the aggregation of proteins and peptides. However, several literature reports in the last decade implicate that single amino acids and non-proteinaceous metabolites also exhibit a tendency to aggregate and form amyloid-like structures [17,18,19,20,21]. Interestingly, the accumulation of metabolites reveals a similar aggregation pathway as that exhibited by conventional amyloidogenic proteins and peptides such as Aβ42, PrP, and α-Syn [22]. Hence, it may be surmised that the patho-physiology of rare inborn errors of metabolisms (IEMs) may have a common etiology for amyloid-associated diseases [22]. The research group of Gazit and coworkers has worked extensively on metabolite assemblies research and reported amyloid-like toxic aggregates formed by single amino acids like phenylalanine, tyrosine, tryptophan, and non-proteinaceous metabolites like uracil, orotic acid, adenine, and oxalic acid, and proposed a “Generic amyloid Hypothesis” which implicates amyloid-like structure formation as a general phenomenon in the patho-physiology of diseases [17,20].

In this review, we will discuss amyloid-like structure formation that occurs in various diseases like AD, PD, PrP diseases, type II diabetes, cancer, and IEMs. Further, we have also discussed in brief functional amyloids and cross-seeding and interaction between different amyloid. It is envisaged that the literature overview presented on this topic will be helpful in understanding the fundamental molecular mechanism behind the origin of several diseases like IEMs, cancer, and type II diabetes and will enlighten the readers about their common etiology with conventional amyloid-associated diseases caused by protein misfolding like Alzheimer’s, Parkinson’s, and a range of prion diseases.

## 2. General Characteristics of Amyloid Structures

The aggregates which are formed by amyloid have some common characteristics like binding to amyloid-specific dyes Congo red (CR) and Thioflavin T (ThT) [23,24]. Typically, amyloidogenic peptides or proteins are rich in β-sheets and hence produce a characteristic negative band around 220 nm in circular dichroism (CD) [25,26]. The morphology of amyloid-like aggregates has been studied by microscopic techniques like transmission electron microscopy (TEM), scanning electron microscopy (SEM), atomic force microscopy (AFM), and light microscopy [18,19,20,21,27,28]. amyloid-like aggregation is commonly associated with the formation of varying morphologies (polymorphs) during the course of disease progression [19]. The early aggregates are usually spherulite-like structures that gradually transform into fibrillar assemblies. The early aggregates, i.e., the pre-fibrillar aggregates of Aβ-peptide, huntingtin, α-synuclein, and transthyretin, are the most toxic and infectious ones [20,29]. These early aggregates impair cellular functions by interacting with cell membranes and causing oxidative stress. Furthermore, pre-fibrillar aggregates increase free Ca^2+^ ion concentration, which eventually leads to apoptotic or neural cell death [30]. Several studies also suggest that not only the pre-fibrillar aggregates but the soluble oligomers of many proteins or peptides are also toxic in major amyloid diseases, for instance, spongiform encephalopathies, Huntington disease, type II diabetes, AD, and PD [31]. Recently, metabolite assemblies have also been characterized using similar methodologies, which insinuate their amyloid character [32].

## 3. Protein Aggregation in Alzheimer’s Disease (AD)

Alzheimer’s disease (AD) is associated with the formation of senile plaques and neurofibrillary tangles in the brain, which causes dementia and memory loss [4]. The plaques are the extracellular deposits of amyloid β (Aβ) protein, while the neurofibrillary tangles are intracellular accumulations [32]. Aβ42 is a peptide formed by 42 amino acids and is the main constituent of the lesions found in the brain of patients with AD, whereas Aβ40 is the most abundant isoform constituted by 40 amino acid sequences [33,34]. Astrocytes and neurons produce Aβ peptides in the brain by the proteolytic processing of the β-amyloid precursor protein (APP) mediated by enzymes such as β-secretase and γ-secretase [35]. It is well-documented that the astrocytes affected by AD express high levels of APP, β-secretase, and γ-secretase, the three main components required for amyloid production [29]. Proteolysis of APP by α-secretase and β-secretase leads to the secretion of sAPPα and sAPPβ, respectively [36]. The secreted sAPPα or sAPPβ have C-terminal fragments which can be cleaved by γ-secretase extracellularly to release the peptides Aβ42 and Aβ40 [36]. Zhao et al. studied the effect of cytokines on the production of amyloids in the astrocytes of the brain in AD-affected patients. Their study suggests that enhanced production of cytokines stimulates the secretion of β-site APP cleaving enzyme 1 (BACE1) in C57BL/6J astrocytes. Furthermore, cytokines strengthen the activity of the β-secretase enzyme in patients suffering from Swedish familial AD mutation [37]. In another study, Calhoun et al. reported overexpression of APP in the mouse model of cerebrovascular disease, cerebral amyloid angiopathy (CAA), and their research suggests that enhanced production of Aβ leads to neuronal loss, microglial activation, synaptic abnormalities, and microhemorrhage in the patho-physiology of CAA [38]. Brothers and others also discovered that secretion of Aβ42 is not only limited to neuronal/astrocyte cells of the brain, but other non-neural tissue such as skin, skeletal muscle, and intestinal epithelium can also secrete Aβ42 [39]. It was also discovered that aggregation of Aβ42 and Aβ40 peptides is caused by a mutation in the gene-producing Aβ peptide [4,40]. Mutations in the *βAPP* (*β*-amyloid precursor protein) gene and two presenilin genes, *PS1* and *PS2*, lead to the production of abnormal Aβ42 peptides, which rapidly aggregate and become deposited as extracellular plaques and tangles in the brain of patients suffering from familial AD [4,40].

Yankner et al. investigated the neurotoxicity produced by Aβ fibrils on hippocampal neuronal cell lines by co-incubation studies under different time periods. Their study suggests that both concentration and period of co-incubation of Aβ40 fibrils with the neuronal cells play a crucial role in predicting the trophic and toxic response of Aβ40 and its effect on neuronal differentiation [41]. In a recent study, Antonino et al. reported the amyloidogenic processing of APP by the oligomers and fibrils of Aβ. It was noted that exacerbated and intracellular accumulation of Aβ42 is caused due to colocalization and physical interaction of APP and BACE1. It was also noted that cells overexpressing the mutant forms of APP, which cannot bind to Aβ, could not increase the colocalization of APP with BACE1, indicating a crucial role of physical binding of Aβ to APP/BACE1 in causing amyloidogenic deposits. Further in this study, it was noted that gallein prevents the Aβ-dependent interaction of APP and BACE1 in endosomes and hence can serve as a good therapeutic target [42]. A comparative analysis of Aβ42 and Aβ40 indicates that Aβ42 has more tendency for oligomerization as well as a more hydrophobic and fibrillogenic nature as compared to that of Aβ40 due to which Aβ42 cause a more cytotoxic response, hence implicating the crucial role of oligomerization in the pathogenesis of neurodegenerative diseases [31]. Aβ42 provides a unique S-shaped triple- or multi-β-sheet motif because of the salt bridge formation between Lys24 and Ala42. Ala42 is absent in Aβ40. The S-shaped triple-β-motif of Aβ42 has Gly29-Ile41 contacts, which further implies that Aβ42 embodies a distinct amyloid strain that has different propagation and structural properties than that of Aβ40 [43].

In 1963, Kidd first reported the structure of amyloid fibrils present in the cerebral cortex of AD patients with the help of electron microscopy (EM). The micrograph revealed that amyloid fibrils exist as paired helical filaments coiled in a squash racket shape [44]. The EM of Aβ42 and Aβ40 mature fibers suggest long straight fibrillar morphology having a diameter in the range of 70–80 Å [45]. The negative-stain EM of protofibrils of Aβ42 and Aβ40 revealed flexible fibers having a diameter in the range of 60–100 Å. The structures of the Aβ42 and Aβ40 protofibrils were also analyzed by AFM, which revealed the diameter of Aβ40 protofibrils to be around 3.1 ± 0.9 nm and the diameter of Aβ40 long fibrillar species as 7.8 ± 0.45 nm [46]. The diameter of Aβ42 protofibrils was found to be 4.2 ± 0.58 nm, while the diameter of bifurcated type-1 and type-2 fibrils of Aβ42 was found to be 7.3 ± 0.53 nm and 3.8 ± 0.43 nm, respectively. Kollmer et al. further reported the cryogenic electronic microscopy (Cryo-EM) structure of Aβ amyloid fibrils from meningeal Alzheimer’s brain tissue. Their studies suggest that Aβ amyloid fibrils have a polymorphic nature, and their morphology is right-hand twisted [47]. The X-ray diffraction (XRD) analysis of amyloid fibers reveals a strong 4.8 Å reflection on the meridian, which corresponds to the hydrogen bonding distance between β-strands. The 10–11 Å reflection on the equator, on the other hand, corresponded to the inter-sheet distance of 10.7 Å [48]. Griffin and his team reported the atomic resolution structure of monomorphic Aβ42 amyloid fibrils. The structure shows that the core of fibril consists of a dimer of Aβ42 molecules, each containing four β-strands in an S-shaped amyloid fold and arranged in a manner that generates two hydrophobic cores that are capped at the end of the chain by a salt bridge [49]. Riek and coworkers presented the 3D structure of Aβ42 fibril, which revealed that residues of 15–42 in Aβ42 form a double horseshoe-like cross β-sheet structure [50]. Circular dichroism (CD) analysis of aqueous solutions of Aβ42 and Aβ40 suggests the aggregates have mostly β-sheet structure [45]. The studies by Soto et al. discussed how α-sheet changes to β-sheet in mutated Aβ42 and characterized this morphological transition using amide shift [51]. A characteristic apple-green color birefringence could also be observed when Aβ42 and Aβ40 aggregates are stained with CR, while an abrupt increase in fluorescence intensity of ThT may be noted when these fibrils were co-incubated with ThT [52]. Several studies have also reported the formation of oligomers as key factors for memory loss and anatomical pathology [53]. The binding studies with oligomeric-specific antibodies proved that an increased level of tau oligomers is found in human AD-affected brains compared to the control [6]. A diagrammatic representation of amyloid formation in Alzheimer’s disease by aggregation of Aβ42 is illustrated in Figure 1 [50,54,55,56,57].

Apart from Aβ peptides, abnormal accumulation and aggregation of the tubulin-associated unit (tau) protein to fibrillar aggregates is another hallmark present in the patho-physiology of AD [58]. The tau protein has six different isoforms generated by alternative splicing, and they are found mainly in the axons of the central nervous system (CNS) [59]. Tau is a phosphoprotein, and its phosphorylation regulates its binding to microtubules. Hyperphosphorylation of tau, which may occur either due to the upregulation of kinase or downregulation of phosphatase, leads to its inability to bind microtubules. Thus, the accumulation of the tau monomer leads to its self-assembly leading to the formation of amyloid-like fibrillar filaments [60]. Negative-stain electron microscopy images of tau inclusions suggest it is composed of paired helical filaments (PHFs) and straight filaments (SFs), which have a fuzzy coat [61]. They have a cross-β structure [62]. Surprisingly, CD studies did not reveal the characteristic negative band at 220 nm for tau, and no secondary structure could be deciphered [62]. Aggregation of the tau protein also leads to many tauopathies apart from AD [58].

## 4. α-Synuclein Aggregation and Parkinson’s Disease

Parkinson’s disease (PD) was reported for the very first time by James Parkinson in 1817. The main pathological characteristics of PD are the formation of Lewy bodies inside the brain [63]. The key components present in the Lewy bodies are misfolded α-Syn peptides [64]. Jiang et al., through their studies, reported mutation in the synuclein alpha (*SNCA*) gene as the main reason behind the abnormal aggregation of α-Syn in PD (Figure 2) [65]. The amyloidosis process of the α-syn peptide includes the formation of soluble oligomers, which further convert into insoluble amyloid fibrillar morphology by self-association [66]. These fibrils are the pathological emblem of PD, and other synucleinopathies are formed due to protein misfolding and aggregation of α-Syn, a 140 amino acid long protein that is normally present at high levels in the brain and is embroiled in crucial synaptic processes in the neurons [67,68]. It is basically a disorganized protein but adopts a partial α-helix motif upon binding with membranes [69]. The presence of the amphipathic region enables the α-Syn protein to bind with membranes [70]. The amphipathic region of α-Syn is followed by the hydrophobic or non-amyloid component (NAC) region, which is mainly responsible for inducing its abnormal aggregation [71]. It was found that β and γ-synuclein are the two isoforms of α-Syn [72]. Dysregulation of cellular processes and toxic gain or losses of function are amongst the major molecular mechanisms that may account for the neurotoxicity associated with α-Syn aggregation [73]. Volles et al. demonstrated the toxicity of α-Syn aggregation through their experiment with synthetic membranes. The results obtained from this study suggest that α-Syn protofibril aggregates cause cell death through the induction of transient cell permeability by the annihilation of vesicle membranes. It is apparent that this transient cell permeabilization may persuade unregulated Ca^2+^ influx, mitochondrial depolarization, and leakage of dopamine in the cytoplasm [74]. Desplats and coworkers, through their experiments, reported α-Syn forms Lewy bodies-like inclusions when transmitted through neurons, ultimately leading to apoptosis and cellular death [75]. Araki et al. proposed PD as a type of amyloidosis. Through immunostaining and XRD studies, they revealed that Lewy bodies (LBs) present in the PD-affected brains contain a cross β-sheet structure [76]. Musteikyte et al. reported the interaction of α-Syn oligomers with the lipid membrane and its implications in the pathogenesis of PD [77]. Dean and Lee identified a link between melanoma and PD since their experiments suggest that aggregation of premelanosome protein 17 (Pmel17) in the melanosome was ameliorated by α-Syn amyloids [78]. A research study by Maji and coworkers reported the crucial role of glycosaminoglycans (GAGs) in the α-Syn amyloid formation. Their experiments suggest that interactions between GAGs and α-Syn promoted amyloid fibrillation, and aggregation was enhanced in a range of proteins by increasing GAG concentration [79]. A recent study by Reyes et al. revealed that α-Syn peptide could accumulate in the liver via originating from the brain in Lewy body diseases. The liver might help to remove the pathological α-Syn aggregates as part of the liver’s detoxification and clearance process [80]. Yamaguchi et al. reported that polyphosphates (polyPs) could persuade α-Syn amyloid formation at both low and high concentrations under neutral pH. At low concentrations, polyPs diminish α-Syn solubility via charge–charge interactions with positively charged N-terminal KTKEGV repeats, whereas at high concentrations, polyPs diminish α-Syn solubility via Hofmeister salting-out effects [81].

Tuttle et al. characterized α-Syn fibrils by using TEM analysis, which revealed that α-Syn fibrils have a width of 4.6 ± 0.4 nm. The X-ray diffraction analysis showed an archetypal meridional diffraction pattern at 4.8 Å, which corresponds to a cross β-sheet structure of α-Syn fibrils. The high-resolution 3D structure of a single untwisted α-Syn amyloid fibril in the substantia nigra of the brains of people who have PD reveals a diameter of around 5 nm as assessed through EM. The structure of a pathogenic fibril of full-length human α-Syn was studied through solid-state NMR and was validated by EM and XRD. These studies suggest that α-Syn fibrils exhibit typical amyloid features, which include parallel, in-register β-sheets and hydrophobic-core residues with substantial complexity arising from diverse structural features like an intermolecular salt bridge, a glutamine ladder, close backbone interactions involving small residues, and several steric zippers stabilizing a new orthogonal Greek-key topology. These characteristics contribute to the robust propagation of this fibril form, as supported by the structural similarity of early-onset PD mutants [82]. Stahlberg and coworkers also elucidated the Cryo-EM structure of α-Syn amyloid fibrils (residues 1–121) at a resolution of 3.4 Å which reveals two protofilaments are intertwined in left-handed helix, and the protofilaments offer Greek-like topology [83]. In another study, Flynn et al. characterized the α-Syn amyloid fibril structure with the help of Raman spectroscopy. Using this spectroscopy, an initial disordered conformation was characterized by obtaining an amide-I bond for α-Syn. Narrowing the amide-I band during aggregation indicates β-sheet structure formation [84]. Kumari et al. proposed a structural insight into α-Syn monomer fibrils. They showed how transient electrostatic interactions drive α-Syn monomer fibril binding with the help of nuclear-magnetic and electron-paramagnetic resonance spectroscopy. Their experimental data presented how the intramolecular unfolding of α-Syn leads to a secondary nucleation process that causes the formation of α-Syn amyloid fibrils [85]. With the help of AFM-IR, Zhou et al. were able to characterize individual α-Syn oligomers. The AFM-IR results revealed that oligomers contain α-helix/random coil, parallel, and antiparallel β-sheet structures in the early stage of aggregation, whereas in the late stage, the parallel β-sheet structure predominates [86]. The CD spectra display a local minimum from 195 to 220 nm, indicating a change in the secondary structure of α-Syn from a mix of α-helix and random coil elements to a β-sheet-enriched structure. The FTIR study also revealed a clear shift of α-sheet to β-sheet structure for α-Syn oligomers [87]. α-Syn like Aβ also binds ThT, which can be noted by enhanced fluorescence intensity of ThT at 480 nm [72]. A diagrammatic representation of amyloid formation in PD is illustrated in Figure 2 [64,88,89,90,91,92].

**Figure 2 life-13-01523-f002:**
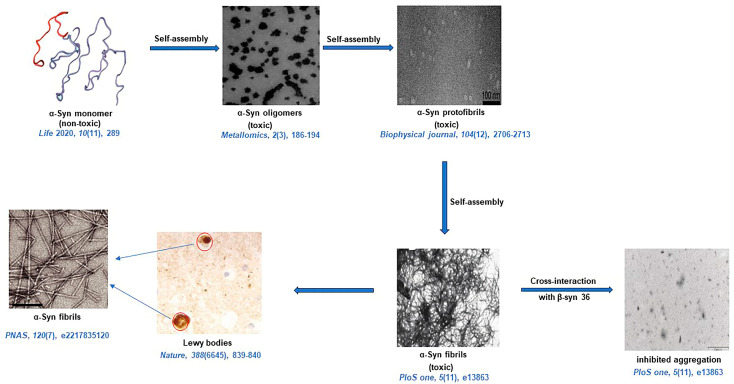
A diagrammatic representation of amyloid formation in PD. Inhibited aggregation is presented by the cross-interaction between α-Syn and β-syn 36 peptide [64,88,89,90,91,92].

## 5. Aggregation of Prion Protein (PrP) and Associated Diseases

Prion is a subclass of amyloid, which is constituted by protein aggregates that are self-perpetuating, self-propagating, and highly infectious, with the capability to cross-seed aggregation in other proteins [93]. Prions and amyloids share a common biochemical basis. However, prions are infectious, whereas other amyloids are non-infectious in higher eukaryotes due to the ease of transmissibility of prion protein from one organism to another. Like amyloids, prions can be pathogenic or functional [94]. Missense, insertion, and point mutations in the prion protein gene (*PRNP*) cause abnormal production of the prion protein (PrP), which has a tendency to self-aggregate and accumulate as insoluble fibrils (Figure 3) [95]. Prusiner et al., for the first time, reported the amyloidogenic nature of prion protein [96]. The main conformational change in the brain during the amyloidosis of PrP is from the cellular form of PrP (PrP^C^) into the disease-causing isoform, i.e., the scrapie form (PrP^Sc^) [97]. Accumulation of prion protein (PrP) amyloids in the brain causes fatal degenerative disorders such as Gerstmann–Straussler–Scheinker disease (GSS), prion protein cerebral amyloid angiopathy (PrP-CAA), scrapie of sheep and goat, spongiform encephalopathy of cattle, kuru, Creutzfeldt–Jakob disease (CJD), and fatal familial insomnia (FFI). PrP^Sc^ is involved with Kuru and Scrapie disease [95]. *PRNP* gene mutation can be an octapeptide repeat insertional mutation [98]. The most common associated mutation of *PRNP* is Glu200Lys [99]. Sanz-Hernández et al. provided molecular insight into the misfolding of human PrP^C^ due to the pathological mutation in T183A [100]. Wang et al. also recently described the E196K mutation in wild-type PrP amyloid fibrils [101]. Goldferb and others clinically investigated the two repeat octapeptide insertions in patients with CJD disease [102]. Cochran et al. studied the five-octapeptide insertional effect in familial CJD patients [103]. Goldfarb et al. carried out research about the transmissible familial CJD disease with five, six, eight, nine, and more extra-octapeptide repeat insertions (OPRI) [104]. Lee and coworkers performed two experiments of PrP variants with octapeptide and without octapeptide insertion. Furthermore, they found that octapeptide repeat insertions affect both folded and misfolded prion proteins. Intriguingly, it was also observed that the deletion of octapeptide forms fewer twisted fibrils, and insertion makes silk-like fibers, but the insertion of octapeptide does not increase cytotoxicity, whereas deletion helps to weaken cytotoxicity [105]. Areškevičiūtė et al. have recently discussed the 8-OPRI in the *PRNP* gene resulting in prion diseases in a Danish family [106]. Different PrP amyloid fibrils can be generated under the same selected experimental conditions [107]. Different types of structures of PrP amyloid fibrils are temperature-dependent and this type of temperature dependency may be due to the size variability in initial PrP molecules [108]. Honda, through in vitro study, showed that Aβ persuades PrP amyloid formation at sub-micromolar concentrations [109]. Artikis et al. examined the accommodation of N-glycans on both ends in each monomer of an octameric parallel in-register intermolecular β-sheet (PIRIBS) protofibril. Through in silico molecular dynamics studies, they proved that the triantennary glycans could be sterically accommodated in register on both N-linked glycosylation sites of each monomer [110]. Takahashi and coworkers reported that PrP-accumulating plaques are associated with Aβ oligomers and appear prior to AD in aged human brains [111]. Sakaguchi and Hara described how the N-terminal region regulates the normal function of PrP^C^, the conversion of PrP^C^ to PrP^Sc^, and the neurotoxicity of PrP^Sc^. The non-structural, flexible N-terminal domain, including the polybasic region, OR region, post-OR regions, etc., regulates the conversion of PrP^C^ to PrP^Sc^ and the neurotoxicity of PrP^Sc^. This specific domain has a role in maintaining the normal function of PrP^C^ [112]. It has been suggested that high temperature and pressure dissociate PrP fibrils at the molecular level [113]. High temperature and pressure-mediated PrP amyloid fibrils dissociation imply that the density of interactions and packing in the amyloid fold is lower than that of the native protein [113]. Purro et al. discussed therapeutic strategies for the interaction between Aβ and PrP^C^ [114]. PrP:Aβ interaction has a crucial role in Aβ-mediated toxicity. It was hypothesized that a fraction of Aβ toxicity is determined by PrP^C,^ which can be therapeutically useful in AD [114].

Molecular mechanisms of PrP-seeded amyloid fibril formation have been deciphered with the help of CD, mass spectrometers, ultracentrifugation, and chemical cross-linking [115]. The TEM study revealed that PrP rods are 10 to 20 nm in diameter and 100 to 200 nm in length in negative staining. By rotating with tungsten, the individual rods are 25 nm in diameter. Cryo-EM analysis evinces that each PrP fibril is composed of two protofibrils intertwined in a left-handed helix, and the fibril core diameter is ~14 nm with a width of ~25 nm [116]. From a solid-state NMR study, it is evident that the PrP^Sc^ fibrils have parallel intermolecular β-sheet architectures [117]. Like other amyloids, prion aggregates also reveal characteristic amyloid dye-binding properties since after binding with ThT, PrP amyloid fibrils exhibited an enhanced fluorescence at 485 nm and revealed apple-green birefringence under cross-polarized light after staining of fibrils with CR [118]. A diagrammatic representation of amyloid formation in prion diseases is illustrated in Figure 3 [116,119,120,121,122].

**Figure 3 life-13-01523-f003:**
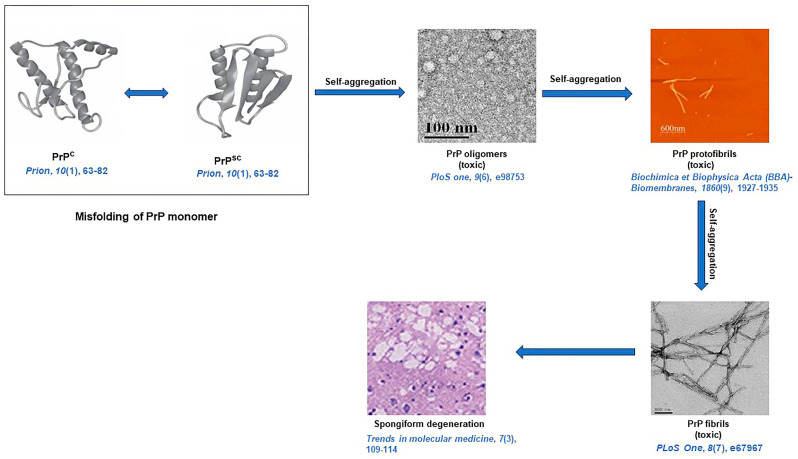
A schematic representation of prion protein (PrP) fibril formation resulting in spongiform degeneration [116,119,120,121,122]. **PrP^C^**: cellular form of PrP, and **PrP^SC^**: infectious form of PrP.

## 6. IAPP Aggregation and Type 2 Diabetes

Islet amyloid polypeptide (IAPP) or amylin is a pancreatic hormone produced by pancreatic β-cells [123]. Amylin is a 32-amino-acid-based peptide. Mutation in the islet amyloid polypeptide (*IAPP*) gene results in the formation of amylin amyloid fibrils, which are primarily observed in late-onset type 2 diabetes (T2D) [124]. In 1993, O’Brien et al. discussed the biological role of IAPP and its function in T2D [125]. Lorenzo and coworkers investigated the toxicity of IAPP fibrils to the β-cells, and from their experiments, it was evident that human IAPP self-assemblies are toxic to islet β-cells [126]. Howard described the deposition of toxic amylin aggregates in primates and cats and correlated those toxic assemblies with T2D in these mammals [127]. Among these mammals, only rats could not produce amyloids, and as a result, they do not generate T2D [128]. Mirzabekov et al. analyzed that human amylin can form ion channels in lipid bilayers, whereas rat amylin cannot form ion channels [129]. Raimundo et al. depicted the role of IAPP amyloids in T2D and AD disease pathology. They explained IAPP-centered drug development strategies against AD as the result of the “diabetes brain phenotype” [130]. Mucibabic and others demonstrated cross-seeding of α-syn and IAPP fibrils in vitro to assess the mutual interaction of two amyloidogenic fibrils on T2D. Their findings state that IAPP amyloid formation is diminished in mice due to the deficiency in endogenous α-Syn. However, when α-Syn is present, β-cells take α-Syn in an IAPP and glucose-dependent manner in vitro. Further, it was also noted that the tail vein-injected α-Syn promotes islet amyloid formation in vivo in mice [131]. Mukherjee et al. proposed that the IAPP amyloid aggregates are just like prion protein aggregates. Their results imply that some of the clinical and pathological changes in T2D might be contagious through a similar manner in which prion proteins disseminate in prion diseases [132]. Kakinen and others demonstrated the coaggregation of IAPP with its primary and secondary amyloidogenic fragments 19–29 S20G and 8–20. They examined that instead of protofilaments, mature fibrils are elongated in coaggregation [133].

Westermark et al., for the first time, investigated the amyloid nature of IAPP aggregates using optical microscopy in T2D [134]. EM, AFM, and X-ray fiber diffraction techniques showed that the structures of IAPP aggregates resemble twisted protofilaments with distinct cross-β sheets. EM of IAPP amyloid fibrils present that the fibrils are long, unbranched, and exhibit a diameter of about 100 Å. An X-ray diffraction pattern depicts that the peak positions of the IAPP fibril occur at 4.7 Å meridional and 10 Å equatorial reflections corresponding to a cross-β pattern. CD and FTIR studies further implicate that the IAPP aggregates predominantly contain β-sheets. The Cryo-EM analysis of full-length IAPP fibrils in ice revealed a strong reflection at 4.7 Å which is similar to a strong meridional signal at 4.7 Å observed for Aβ (11–25) fibrils, supporting this cryo-EM feature as a common characteristic present in amyloid fibrils [135]. Cooper et al. monitored the growth, bi-directionality, and morphological change of individual amylin fibrils using time-lapse AFM. During the growth of a protofibril, it is elongated at both ends, indicating a bi-directionality of protofibril growth [136]. The structure of protofibril of amylin suggests that about 2.6 human amylin molecules are packed in 1nm protofibril [137]. Roeder and coworkers described the similarities of IAPP fibrils with amyloid-β fibrils with the help of Cryo-EM [138]. Their atomic model reveals two S-shaped fold-intertwined protofilaments of IAPP in the main polymorph that resemble the S-fold of Aβ fibrils in AD [134]. In the solid-state NMR (ssNMR) study, β-hairpin structures are observed in IAPP amyloid fibrils, and this β-hairpin structure consists of two β-strands [139]. Like other amyloids, after staining with CR, apple-green birefringence was observed under cross-polarized light, and enhancement in fluorescence intensity of ThT was noted after binding of IAPP aggregates [135]. A diagrammatic representation of amyloid formation in type II diabetes is illustrated in Figure 4 [140,141,142,143].

## 7. Amyloid-like Aggregation in the Patho-Physiology of Cancer

The p53 protein acts as a global transcription factor to maintain the integrity of the cells under different stresses [144]. The p53 consists of 393 amino acid residues and abnormality in its biological function leads to cancer. The mutation in the tumor-suppressor gene (*TP53*) causes the formation of abnormal p53 protein, which has a tendency to self-aggregate and form amyloidogenic structures [145,146]. Interestingly, the aggregation propensity of the p53 protein is remarkably like the PrP protein, and likewise, PrP and p53 aggregates are also infectious and cause cross-seed aggregation in other proteins [146]. Maji and his team, through their studies, demonstrated that aggregation of p53 protein results in the impairment of its regular functions leading to the production of cells with leaky membranes, a characteristic feature of cancer pathogenesis [147]. Several research findings around the globe suggest that almost half of human cancers are associated with p53 gene mutation [145,147]. The studies by De Oliveira and others suggest that the p53 DNA binding domain (DBD) is more prone to pressure-induced unfolding and aggregation due to poorer backbone hydrogen bond formation between p53 and DBD [148]. Maji and coworkers examined the contribution of wild-type p53 aggregates in the formation of tumors and hypothesize its implications as potential therapeutic targets [149]. Maritschnegg et al. detected p53 aggregates using the ELISA system (enzyme-linked immunosorbent assay) and further enhanced the understanding of the influence of p53 misfolding in cancer. The Seprion-ELISA system binds with high molecular p53 aggregates, whereas tetramers of p53 do not bind with the Seprion-ELISA system [150]. Farmer et al. reported the formation of p53 oligomers and fibrils in human AD brains. Furthermore, their studies implied that in AD, impairment of DNA might occur due to the interaction of p53 oligomers with tau oligomers [151]. Maji and his group carried out circular dichroism (CD) analysis of p53 amyloids, which revealed a β-sheet-rich character of p53 aggregates with a negligible fraction of α-helix. From the TEM study, both fibrillar and non-fibrillar structures of p53 could be observed. Further, like other amyloid, p53 aggregates could also bind ThT and CR dyes [147]. Solution state that NMR evinces disruption of the salt bridge between monomers during p53 tetramerization as the reason which leads to oligomerization and amyloid-like fibril formation by p53 [152]. A diagrammatic representation of amyloid formation by the p53 protein in cancer is illustrated in Figure 5 [148,153,154].

## 8. Metabolites Assemblies as a Surprising Extension to Generic Amyloid Hypothesis

### 8.1. Amyloid-like Aggregation of Single Amino Acids and Its Implications in IEMs

Gazit and coworkers, for the very first time, delineated a common etiology between rare IEM phenylketonuria (PKU) and amyloid-associated diseases. In their pathbreaking research, the group reported amyloid-like fibrils formed by the self-assembly of phenylalanine (Phe). Mutation in the phenylalanine hydroxylase (*PAH*) gene causes deficiency in the phenylalanine hydroxylase (PAH) enzyme. The blood concentration of Phe is elevated due to the deficient activity of the PAH enzyme. Elevated Phe concentration is responsible for autosomal recessive disorder PKU. The study revealed that Phe fibrils, just like conventional amyloid, could bind CR and ThT dye. The MTT assay suggests that Phe fibrils are cytotoxic to both hepatic as well as neural cell lines. The immune-histochemical analysis connotes the formation of antibodies against Phe fibrils. Hence, the results illustrated in this work suggest a common etiology between PKU and amyloid-associated diseases and implicates that pathogenesis in PKU is associated with the formation of toxic fibrillar assemblies by phenylalanine [17]. Concomitantly, the same research group investigated the morphological, structural, and mechanical properties of Phe nanofibrils in both hydrated and dehydrated states by using AFM and electron microscopy techniques. Their report described that Young’s modulus of Phe fibrils might show a value of up to 30 GPa, which is higher than any other biological fibrillar structures, revealing their exceptional mechanical strength [155].

Subsequent to the Phe fibril discovery, Gazit and his team explained the abrupt formation of amyloid-like supramolecular nanostructures by the self-assembly of tyrosine (Tyr) [20]. The rare IEM disorder tyrosinemia occurs due to the accumulation of Tyr in the body, which is also caused by a mutation in the gene-expressing enzyme tyrosine hydroxylase (TyH), which catabolizes Tyr [156]. Ménard-Moyon et al. encountered that the Tyr amyloid nano-fibrillar structures accelerate the aggregation of globular proteins and aromatic metabolites, and this type of amyloid cross-seeding results in the creation of a lethal aggregation trap of proteins. For instance, tyrosine can self-assemble into a variety of structures, such as nanoribbons, dendritic structures, fiber-like structures, etc. [157]. Tyr amyloid structures display an apple-green color after staining with CR and an enhanced fluorescence intensity after binding with ThT. The cytotoxicity analysis of Tyr assemblies by XTT assay suggests that upon increasing the concentrations of Tyr from 0.2 to 4 mg/mL, the cell viability was decreased from 80% to 40% [20]. Further, Gazit and coworkers developed anti-Tyr antibodies to inhibit the formation of toxic Tyr amyloid fibrils. Their study revealed that pre-incubation of Tyr assemblies with anti-Tyr antibodies resulted in almost 80% cell viability in SH-SY5Y cells, suggesting a possible therapeutic cure for tyrosinemia by these antibodies [158]. Further experimental studies by Kar and coworkers elucidate the ability of Tyr assemblies to cross-seed amyloid-like aggregation in lysozyme, BSA, and myoglobin proteins under physiological conditions [159]. Subsequently, Gazit and coworkers also reported toxic amyloid-like fibrils formed by the aggregation of tryptophan (Trp), which induces apoptosis in neuronal cells and has implications in hypertryptophanemia and Hartnup diseases [160]. MD simulation experiments for assessing the mechanism of cytotoxicity caused by Trp and Phe fibrils suggest that these metabolite assemblies can penetrate inside the membrane, leading to its disruption and ultimately inducing apoptosis [161].

Although the aggregation of aromatic amino acids to amyloid-like morphologies was well-studied by Gazit and coworkers, amyloid-like aggregation propensities for non-aromatic amino acids were unknown. In this context, our research group, for the very first time, reported amyloid-like structures formed by non-aromatic amino acids cysteine (Cys) and methionine (Met). The amyloid-like aggregation of Cys and Met was validated through microscopic techniques like SEM and TEM and spectroscopy tools like solid-state NMR, FTIR, TGA, and XRD, along with preliminary MD simulation studies. The MTT assay also revealed that both Cys and Met fibrils are cytotoxic and decreased cell viability both in neural and kidney cells, suggesting the role of amyloidogenic pathways in diseases like cystinuria and hypermethioninemia [18]. Further, our group reported unusual aggregates formed by proline, hydroxyproline, and lysine [19]. Currently, we are also studying the aggregation properties of non-aromatic polar amino acids, and our initial experiments suggest that glutamine (Gln), aspartic acid (Asp), and glutamic acid (Glu) may aggregate to amyloid-like structures [162]. Self-assembly of glycine (Gly) crystalline fern-like architects are also reported [163]. Gazit and coworkers have also previously studied the self-assembly of cystine to amyloid-like fibrillar morphologies using TEM, confocal fluorescence microscopy, and ThT-binding assay. It was also discovered that cystine aggregates are cytotoxic and induce cell death up to 62%, as indicated by annexin V and propidium iodide (PI) apoptosis assay [20]. Gazit and coworkers illustrated well-organized supramolecular β-sheet architects of all naturally occurring amino acids at the nano level. Hence, this study suggests a potential for self-assembly in all naturally occurring amino acids, which needs to be deciphered in future studies [164]. A diagrammatic representation of amyloid-like structure formation by single amino acids is illustrated in Figure 6. Plasma concentrations of all amino acids are also mentioned in Table 1.

### 8.2. Non-Proteinaceous Metabolite Assemblies and Their Implications in IEMs

Gazit and coworkers have recently reported research on the association of homocysteine amyloid toxic fibrils with Alzheimer’s disease pathogenesis. They characterized the homocysteine crystals and their amyloid fibrils. Furthermore, they demonstrated the inhibition methodology of homocysteine fibrils by polyphenolic compounds [165]. Zaguri et al. examined the self-assembly process of oxalate, which results in supramolecular nanofibrils without the accumulation of calcium. They provided mechanistic insight to explain the inconsistency between impaired retinal function and lack of crystal deposition by examining the self-assembly of oxalate into supramolecular nano architects. Using TEM analysis, they displayed the existence of elongated oxalate nano-fibrillar structures. They also obtained the crystals of oxalate by using a supersaturated solution containing ~70 mM of calcium oxalate in phosphate buffer saline (PBS) [166]. Quinolinic acid (QA) is a neurometabolite in the kynurenine pathway, the biosynthetic pathway of tryptophan, which is associated with deadly neurodegenerative diseases. Tavassoly et al. first demonstrated the self-assembled nanostructures formed by QA. They showed the QA self-assembly by dissolving QA in PBS at 90 °C to obtain a homogenous solution followed by gradual cooling of the solution followed by the characterization of the morphologies formed by TEM. From TEM micrographs, it was observed that the fibrillar aggregates are homogeneous in length and width and have diameters between 3 and 6 nm. They further studied the cytotoxicity and apoptotic behavior of QA fibrils on human neuroblastoma cells. Surprisingly, it was observed that QA assemblies can result in up to 100% cell death. Further, it was noted that QA nanofibrils can be disrupted by EGCG. To examine the cross-seeding effect of QA fibrils into the α-Syn self-aggregation, they co-incubated α-Syn with QA seeds, and after a couple of hours, it was found that those QA seeds induced α-Syn aggregation [167].

Gazit and his team delineated the cytotoxicity of adenine, orotic acid, and uracil self-assemblies. Cytotoxicity of adenine was carried out using an XTT assay in SH-SY5Y cells, and it was found that as the concentration of adenine was increased to 2 mg/mL, cell viability decreased around 50%, which is similar to that of orotic acid. Uracil assemblies, on the other hand, displayed 90% cell viability at the same concentration. From the cytotoxicity analysis, it was surmised that adenine and orotic acid self-assemblies exhibit more cytotoxicity compared to uracil aggregates. On the other hand, the apoptotic behavior of adenine, orotic acid, and uracil assemblies was analyzed using annexin V and propidium iodide (PI) assays. This suggests that adenine and uracil assemblies cause from 40% to 50% cell apoptosis, whereas orotic acid exhibits about 30% cell apoptosis [20]. Gazit and coworkers also illustrated the adenine fibril formation and its accumulation in a yeast model [168].

Gazit and coworkers recently reported robust twisted ribbon-like structures formed by the self-assembly of glucosylceramide (GlcCer). Notably, a deficiency of the lysosomal enzyme glucocerebrosidase results in the accumulation of GlcCer and causes Gaucher’s disease [169]. The cytotoxicity analysis of GlcCer aggregates, along with other characteristic amyloidogenic traits, suggest that the etiology of Gaucher’s disease is also associated with amyloidosis [169]. Kar and coworkers discussed amyloid-mimicking self-assemblies formed by artificial sweetener aspartame through conventional microscopy analysis, turbidimetry, ThT-binding assay cytotoxicity, and hemolysis analysis, along with MD-simulation experiments [170]. Further, they also recently reported amyloid-like structures formed by dopamine using similar methodologies [171]. In our recent research, we have extended the generic amyloid hypothesis to assemblies formed by the metabolites of the urea cycle and uric acid pathway [172]. A diagrammatic representation of amyloid-like structure formation by non-proteinaceous metabolite is illustrated in Figure 6 [17,26,169,173,174].

**Figure 6 life-13-01523-f006:**
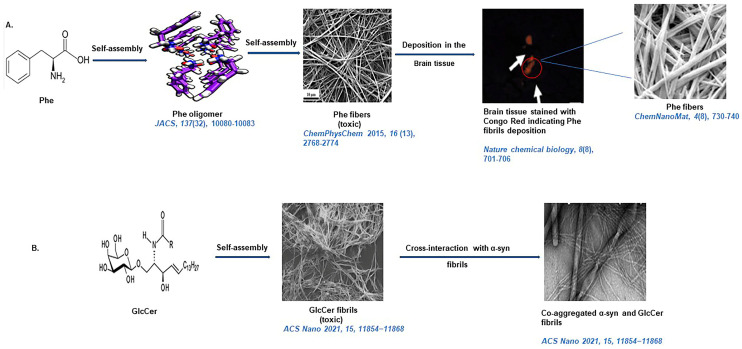
A diagrammatic representation of amyloid-like structure formation by proteinaceous and non-proteinaceous metabolites. Upper panel (**A**): amyloid-like structure formation by phenylalanine (Phe). Lower panel (**B**): amyloid-like fibrils generated by glucosylceramide (GlcCer) and its interaction with α-Syn fibrils [17,26,169,173,174].

## 9. Functional Amyloid

We have discussed in this review different types of amyloid that cause pathogenesis in several diseases. However, amyloids are not always harmful, albeit they also play important functional roles in nature, like adherence to abiotic surfaces, resistance to antibiotics, detoxification of harmful compounds, assisting in electron transport, morphological differentiation of filamentous bacteria, and biofilm formation. Although the term amyloid is often associated with diseases, scientists have found various cases where amyloid fibrils play important functional roles in nature, from prokaryotic cells to mammals. Amyloids help to adhere to abiotic surfaces [175], detox harmful compounds [176], resist antibiotics [177], direct morphological differentiation of filamentous bacteria [178], and assist electron transport [179]. Many bacteria express and secrete amyloid proteins to the extracellular environment, which, upon self-assembly, form biofilm [180]. These functional roles of amyloid occur in nature from prokaryotes to mammals. In bacterial strain *E. coli*, CsgA proteins assemble on top of the nucleator protein CsgB to form amyloid-like fibrils, which form biofilm [181]. The eggshell of silk moths is also made of amyloid-like fibrils that have exceptional mechanical strength [182]. In mammalians, functional amyloids play crucial roles in long-term memory formation and hormone storage and reduce the toxicity of melanin formation by minimizing the diffusion of highly reactive toxic melanin precursors out of melanosomes [183,184,185]. Hence, these findings suggest that amyloids also have a beneficial role in physiology.

Functional amyloids are also excellent candidates for the design of hydrogels. Natural amyloids formed by β-lactoglobulin and lysozyme proteins have been used for the fabrication of antibacterial hydrogels. The mechanical properties of amyloid fibrils are better than actin or tubulin, owing to their higher Young’s modulus of 3.3 ± 0.4 GPa and a peak tensile strength of 0.6 ± 0.4 GPa [186,187]. Hence, amyloid fibers are excellent scaffolds for the design of structural materials like free-standing films and fibers [188,189,190]. Amyloid fibrils have also been used for the design of organic–inorganic hybrid nanomaterials owing to their high aspect ratios and the presence of multiple binding sites along their surface, which allows the fabrication of nanoscale amyloid-inorganic hybrid materials through post-assembly or co-assembly modification [191]. Many macroscopic composite materials have been designed using amyloid fibrils. Further, modification of amyloid fibrils with unique functional components can also render them responsive to pH, temperature, salt concentration, magnetic field, or other environmental stimuli. Hence, many literature reports suggest potential applications of macroscopic amyloid materials for the fabrication of stimuli-responsive sensory materials. Conductive wires and gels, which are biocompatible and biodegradable, have also been designed by amyloid fibrils owing to their highly ordered structures which facilitate high electron mobility. Further, engineered amyloid fibrils with an incorporated catalytic site also provide unique chemical environments to catalyze reactions making them immensely useful for catalysis. Thus, amyloids are excellent candidates for the design of functional materials and have numerous applications, such as hydrogels, fibers, composites, sensors, and catalysts.

## 10. Amyloid Cross-Seeding and Interaction

The formation of amyloid fibrils is regulated by the formation of “seeds”, which are stable nuclei that promote fibril formation by converting soluble proteins to fibrils. Amyloid fibrils formed by aggregation of specific proteins are often entangled and found in association with other proteins. For example, neurofibrillary tangles present in AD have both Ab and tau fibrils. Hence, cross-seeding or cross-talk between different amyloidogenic fibrils is a common phenomenon that may be homologous or heterologous [192]. In homologous cross-seeding seeds of one type of protein interact and facilitate fibril formation, while in heterologous cross-seed, the seed of one type of protein facilitates the aggregation of another amyloid protein. Hence, more than one misfolded protein can cause pathogenesis in one disease, and more than one type of protein misfolding disease (PMD) can co-exist in an individual [193,194]. It has also been observed that individuals diagnosed with one PMD are more susceptible to developing another [195,196]. There is a plethora of literature reports wherein cross-seeding of known amyloidogenic proteins, namely Aβ42, Aβ40 PrP tau IAPP, and α-Syn, has been cross-seeded with each other, and the results illustrate that cross-seeding has a synergistic effect on the aggregation propensities and toxicities associated with individual amyloidogenic proteins [197]. Hence, to study the effect of cross-seeding in the pathogenesis of Alzheimer’s diseases, for instance, Aβ42/Aβ40 proteins are cross-seeded with variants of PrP, IAPP, tau, and α-Syn, which suggest these proteins impel Aβ-associated toxicities.

Cross-interactions might also have beneficial roles when the presence of other proteins inhibits the formation of toxic aggregates by major amyloid constituents. Some proteins, for example, TTR, CysC, and apoA-1, inhibit Aβ fibrillation, leading to delayed AD progression [193]. Many small molecule inhibitors have also been efficiently identified by cross-seeding experiments. In this context, a small molecule, MG-2119, was screened as a potent inhibitor for both monomeric tau and α-Syn aggregation [198]. The oligomerization and fibrillogenesis of both Aβ and α-Syn protein can be efficiently inhibited by anti-Parkinsonian drugs entacapone and tolcapone [199]. Aggregation of Aβ and α-syn can also be inhibited by curcumin [200]. Fibrillation of both Aβ and hIAPP can be inhibited by polyphenol pentagalloyl glucose (PGG) [201,202]. Similarly, aggregation of both Aβ and tau can be inhibited by epigallocatechin-3-gallate (EGCG), a polyphenol constituent of green tea [203]. Indeed, polyphenols have been identified as generic amyloid inhibitors for all amyloidogenic protein sequences as well as metabolite amyloid [204,205]. Immunotherapy is another approach whereby antibodies are designed against amyloid antigens, and, in this context, anti-prion, anti-syn, and anti-A have been identified by cross-seeding.

## 11. Critical Analysis and Future Outlook

The studies illustrated in this review provide a comprehensive literature survey that implicates the validity of the generic amyloid hypothesis in the etiology of a plethora of diseases like AD, PD, prion diseases, T2D, cancer, and IEMs. Abundant evidence also suggests that common therapeutic remedies for these diseases can be envisaged by designing drugs that could act as generic amyloid inhibitors. However, it is apparent that the diseases like AD, which was first discovered to be caused by amyloid-like aggregation, still do not have a cure. This is surprising, and the reason for the inefficacy of these drugs is still unknown. Hence, if the generic amyloid hypothesis is considered a common cause for pathogenesis in other diseases, including the most recently discovered association with IEMs, its significance in unraveling the therapeutic remedy for these diseases is still questionable since a drug that inhibits such amyloid-like progression should have been clinically successful. However, the limited success of such drugs in AD implies that there might be another pathway through which the disease may progress. The clinical trials suggest that most drugs that could inhibit aggregation and hinder amyloid-like aggregation have not been found effective in the treatment of this disease. There should be a lot more clinical studies for assessing the effect of generic amyloid inhibitors like polyphenols, tannic acid, quercetin, and flavonoids as common therapeutic drugs for the diseases mentioned in the review to unravel the role of the amyloid hypothesis in their etiology. Considering the failures of previous drug trials, it is imperative to design a new drug or screen a potential FDA-approved drug that could arrest aggregation at the nucleation step of amyloid growth. One possible reason for drug failures might be a lack of understanding of therapeutic targets or identification of wrong pathological substrates. Hence, it is imperative to pursue more research on the ultrastructure of different amyloids and understand the mode of their propagation and transmission. Besides drugs, immunotherapy can play an efficient role in targeting these amyloidogenic antigens and may offer long-term immunity against these diseases, alleviating the harmful effects of drug administration that yield short-term results. Some anti-Aβ antibodies have shown promising results by crossing the blood–brain barrier, leading to the dose and time-dependent reduction in Aβ. Anti-prion antibodies are also available, and studies suggest they could lower the scrapie prion protein level and protect the neurons against the infectious prion protein [206]. However, evidence is also accumulating that the development of anti-Aβ antibodies and anti-prion systems has been filled with confusing and disappointing results since they are set too late in the disease progression and severity [207]. Hence, it is essential to administer such antibodies at the very early stage of the disease. Coupling anti-prion systems with other anti-amyloid antibodies might also be effective depending on how they interact with the neurotoxic conformations, i.e., oligomers and protofibrils. Further, studies should also be conducted on various morphological transitions which occur in amyloid formation and the effect of toxicity of aggregates present in different stages, right from nucleation to prefibrillar aggregation and fibril formation. It might be possible that the drugs which inhibit amyloid do not work as the aggregates, which are formed in the preliminary nucleation step, might be more lethal/toxic. Moreover, most of the studies wherein generic amyloid hypotheses have been proposed are performed only at in vitro stage. Still, there is an urgent need for extensive in vivo studies that may pave the way for clinical trials, which appear to be a far-reaching goal at present. Table 2 summarizes the characteristic features of different types of amyloids associated with the pathogenesis of diseases discussed in the review for a quick revision of the readers.

## 12. Conclusions

Overall, in this review, we have discussed a common etiology for diseases like Alzheimer’s, Parkinson’s, prion diseases, type II diabetes, cancer, and IEM from an amyloid perspective. The studies illustrate that there is a common origin of these diseases, i.e., at first, a mutation in the gene causes enzyme dysfunction, which further leads to the accumulation of protein/metabolite. These accumulates have a tendency to self-assemble and form amyloid-like aggregates inside the body, which in turn produce cytotoxicities that manifest in the form of disease. In this review, we have tried to present a comprehensive literature overview of the major studies completed to understand the process of amyloid formation and its implications on the patho-physiology of associated diseases. These studies suggest a crucial role of amyloids in the etiology of diseases and also illustrate them as a potent therapeutic target for the treatment of these diseases. In this context, cross-interaction that may occur among different amyloidogenic proteins/metabolites needs to be studied in greater detail in the future to understand the implications of the coexistence of more than one amyloid in an individual. There is also a greater need to do more in vivo studies to understand the physiological concentrations and conditions in which amyloid formation occurs. More research also needs to be pursued in isolation of amyloidogenic aggregates present in living organisms and understand their structure and conformation in a detailed manner. The oldest known amyloid-associated disease, i.e., Alzheimer’s, still does not have a cure. This is a serious matter of concern, and effective drugs which could arrest the self-association of monomers of these amyloidogenic proteins/metabolites are urgently required. Hence, more cross-seeding studies should be pursued to screen FDA-approved drugs which could inhibit amyloid formation at the earliest stage. In this context, immunotherapy may also play an important role, and antibodies can be engineered against these amyloidogenic antigens. It is encouraging that the Anti-PrP and Anti-Syn antibodies are now commercially available. To find a therapeutic remedy against any disease, it is necessary that its patho-physiology should be understood precisely, and a therapeutic target should be identified. We believe that more extensive research will be pursued to unravel the role of amyloidosis in the pathogenesis of diseases mentioned in this review. Such studies can be useful in paving the way for identifying a common/generic therapeutic module for the treatment of these maladies in the future.

## Figures and Tables

**Figure 1 life-13-01523-f001:**
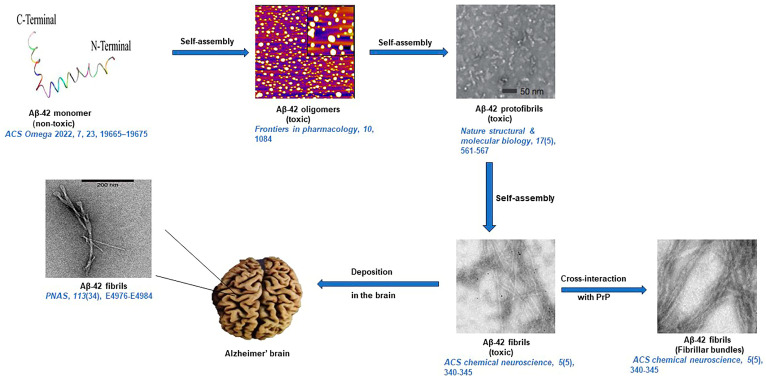
A diagrammatic representation of the amyloidogenic toxicity mechanism of Aβ42 fibrils’ formation and deposition in Alzheimer’s brain. A cross-interaction is shown between Aβ42 fibrils and prion protein (PrP) [50,54,55,56,57].

**Figure 4 life-13-01523-f004:**
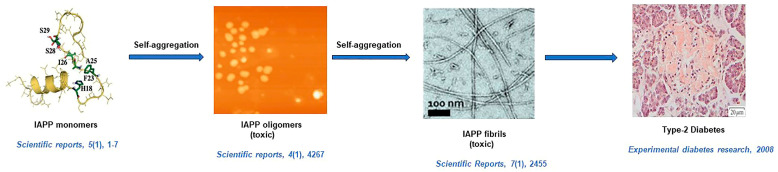
Mechanism towards amyloid fibril formation in type II diabetes. Inhibited aggregation is shown by the cross-interaction between IAPP and lysozyme (Lys) [140,141,142,143].

**Figure 5 life-13-01523-f005:**
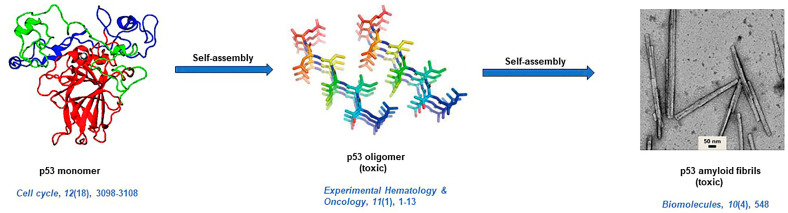
Mechanistic pathway of p53 amyloid fibril formation and its deposition in lung tissues [148,153,154].

**Table 1 life-13-01523-t001:** Amino acids and their plasma concentrations.

Amino Acids	Plasma Concentration Range (µmol/L)
Alanine	564–644
Arginine	44–51
Asparagine	92–98
Aspartate	64–69
Glutamate	261–296
Glutamine	506–547
Glycine	390–452
Histidine	114–122
Isoleucine	96–102
Leucine	191–210
Lysine	210–242
Methionine	27–31
Proline	244–265
Phenylalanine	95–101
Serine	191–201
Tyrosine	73–83
Threonine	164–168
Tryptophan	65–72
Valine	217–233

**Table 2 life-13-01523-t002:** Amyloids and its characteristics.

Type of Amyloids (Disease Caused)	Aβ42/40 (AD)	tau (AD)	α-Syn (PD)	PrP (Prion Diseases)	IAPP (Type 2 Diabetes)	p53 (Cancer)	Metabolite Amyloid (IEMs)
**Characteristics**	1. Aβ42: 42 amino acids-based peptide. Aβ40: 40 amino acids-based peptide. 2. Mutations In *βAPP*, *PSN1*, and *PSN2* genes. 3. Accumulation of neurofibrillary tangles and plaques. 4. Disruption in cell membrane.	1. Six tau isoforms of varying length present. Longest human CNS tau isoform with 441 residues is chosen as model to exemplify tau primary structure. 2. Mutations in *mapt* gene. 3. Hyperphosphorylation of tau leads to its inability to bind microtubules and excess tau self-assembles to form toxic tau oligomers and fibers.	1. 140 amino acid-based long protein. 2. Mutation in *SNCA* gene. 3. Accretion of α-Syn. 4.Formation of Lewy bodies. 5. Disruption in cellular membrane.	1. 208 amino acid-based protein. 2. Isomeric forms of PrP: cellular form: PrP^C^ and scrapie form: PrP^Sc^. 3. *PRNP* gene mutation. 4. *PRNP* gene can be missense, insertion, or point mutations. 5. Prion protein accumulation. 6. Disruption in cell membrane.	1. 37 amino acid-based polypeptide. 2. Regulatory peptide in the islets. 3. Inhibits insulin and glucagon secretion. 4. Human plasma IAPP concentration is only 1–2% of that of insulin. 5. Mutation in the *IAPP* gene. 6. This amyloid causes type 2 diabetes. 7. Disruption in cell membrane.	1. 393 amino acid-based protein. 2.*TP53* gene mutation. 3. Accumulation of p53. 4. p53 is a tumor-suppressor protein. 5. Disrupts cell membrane.	1. Few amino acids and non-proteinaceous metabolites exhibit amyloid. 2. Mutations in specific genes. 3.These amyloids can induce cross-seeding. 4.These amyloids can act as functional amyloids. 5. These amyloids cause rare inborn errors of metabolism. 6. Cell membrane disruption.
**Secondary** **Structure**	β-sheet	Cross β-sheet	β-sheet	β-sheet	β-sheet	β-sheet	β-sheet
**X-ray**	4.8 Å meridional and 10–11 Å equatorial reflections.	Reflection at 4.7 Å	4.7 Å meridional and 10 Å equatorial reflections.	4.6–4.7 Å meridional and 8.5–9.7 Å equatorial reflections.	4.7 Å meridional and 10 Å equatorial reflections.	4.7 Å meridional and 10 Å equatorial reflections.	Predicts crystalline or non-crystalline nature in assembly process.
**CD**	In case of Aβ40, majority portions are β-sheet structures, whereas Aβ42 contains only β-sheet structures.	No secondary structure could be identified in CD.	A mix of α-sheet and random coil to β-sheet-enriched structure.	Highly ordered β-sheet conformation.	IAPP aggregates contain mostly β-sheet.	Both fibrillar and ordered p53 aggregates have β-sheet rich profile.	β-sheet structure.
**CR**	Apple-green birefringence color.	Apple-green birefringence color.	Apple-green birefringence color.	Apple-green birefringence color.	Apple-green birefringence color.	Apple-green birefringence color.	It might display apple-green birefringence color.
**ThT**	ThT fluorescence intensity at 480 nm.	ThT fluorescence intensity at 485 nm.	ThT fluorescence intensity at 480 nm.	ThT fluorescence intensity at 485 nm.	ThT fluorescence intensity at 480 nm.	ThT fluorescence intensity at 480 nm.	It may display an enhanced intensity at 480 nm due to ThT fluorescence.
**Location**	Brain	Brain	Brain	Brain	Pancreatic β-cells	Cancer tissues	Brain and other organs

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
