# Peer review of "Realization of Amyloid-like Aggregation as a Common Cause for Pathogenesis in Diseases"

_life, 2023, doi:10.3390/life13071523_

Round 1

Reviewer 1 Report

In this review, the authors made an overview about the general characteristics of amyloid aggregates of the most common amyloid proteins, linking their mechanism of self-assembly to the aggregation process of metabolites, thus offering an hypothesis of a common cause for the pathogenesis in diseases. The review and the idea are well presented and organized, however some concerns come in mind during the reading.

1. The choice of the amyloids, here presented, is not completely clear? Did the authors decide to show only exemples of extracellular amyloid proteins? Why did not present tau protein too? 

2. In most of the paragraphs, the authors made some exemples of amyloid cross-interaction. This is really interesting and innovative pointof view in the field. I recommend to make a general paragraph concerning this cross-interactions, by explaining particularly the positive and negative impact of these kind of interactions.

3. I suggest to revise all the figures. The representations are too much simplistic. They need a common design, for example including the morphology of the fibrils and oligomers even at atomic level, the mechanism of toxicity and which aggregate is known to be toxic, the possible cross-interaction and finally the optical microscopic images of aggregates.

4. Sentence lines 536-537: this conclusion has been made so fast. There are some hypotheses explaining the inefficiency of drugs, such as lack of knowledge about the real toxic species, the absence of an early diagnostic tool and the absence of information about the real concentration that should be reached to have a therapeutic effect into the brain. Please explain better.

5. Sentence lines 548-551: this is true for amyloids like alpha synuclein, tau etc. This is quite difficult to follow for IEMs. Please provide a better link between the common known amyloid aggregationn and the one observed for metabolites. In one case, proteins and peptides with a well-defined structure are involved, in the other only small molecules. Please explain better this concern and provide a better link between traditional amyloidoses and IEMs.

6. The table 2 is difficult to read. Please reorganize the table by summarizing the results and not providing full sentences. Please pay attention to typography. In the X-ray part don't forget to provide the pdb code.

7. In the conclusion section, the authors should be carefull and not give inappropriate considerations. A common etiology or a common origin between amyloidoses and IEMs are not the correct conclusions. This might be true for the hereditary forms but for the others? The sentence lines 564-571 is too long and difficult to understand.

Minor suggestions:

- line 72: is it not 220 nm?

- paragraph 3: don't forget to explain the difference between Abeta 1-40 and Abeta 1-42 by citing the work of Xiao et al. where a three-strand beta sheet structure of Abeta 1-42 is provided as possible explanation of the fast aggregation propensity.

- line 173: a full stop after the alpha letter

- line 231: space

- line 356: is it abeta 1-4à or 1-42? In the brakets are the numbers of residues?

- line 386: two times the ref 126

- lines 451-469: organisation of the paragraph.

- Table 2 line 473: it should be table 1 and it should be present a link with the text. It is not commented.

English language needs some minor editing.

Author Response

Response to Reviewer 1 comments to the authors:

Reviewer 1

In this review, the authors made an overview about the general characteristics of amyloid aggregates of the most common amyloid proteins, linking their mechanism of self-assembly to the aggregation process of metabolites, thus offering a hypothesis of a common cause for the pathogenesis in diseases. The review and the idea are well presented and organized, however some concerns come in mind during the reading.

Comment 1. The choice of the amyloids, here presented, is not completely clear? Did the authors decide to show only examples of extracellular amyloid proteins? Why did not present tau protein too? 

Response: Thank you for pointing this out! We showed examples of only extracellular amyloids in the review. As per reviewer suggestion we have now also discussed in brief aggregation of Tau protein.

Comment 2. In most of the paragraphs, the authors made some examples of amyloid cross-interaction. This is really interesting and innovative point of view in the field. I recommend to make a general paragraph concerning this cross-interactions, by explaining particularly the positive and negative impact of these kind of interactions.

Response: Thank you for this suggestion. We have added a separate section about cross seeding in revised draft and it is indeed an interesting phenomenon.

Comment 3. I suggest to revise all the figures. The representations are too much simplistic. They need a common design, for example including the morphology of the fibrils and oligomers even at atomic level, the mechanism of toxicity and which aggregate is known to be toxic, the possible cross-interaction and finally the optical microscopic images of aggregates.

Response: Thank you for this valuable assessment. We have modified all the figures. We have also reduced the number of figures from 7 to 6.

Comment 4. Sentence lines 536-537: this conclusion has been made so fast. There are some hypotheses explaining the inefficiency of drugs, such as lack of knowledge about the real toxic species, the absence of an early diagnostic tool and the absence of information about the real concentration that should be reached to have a therapeutic effect into the brain. Please explain better.

Response: Thank you for the comment. We modified the conclusion section.

Comment 5. Sentence lines 548-551: this is true for amyloids like alpha synuclein, tau etc. This is quite difficult to follow for IEMs. Please provide a better link between the commonly known amyloid aggregation and the one observed for metabolites. In one case, proteins and peptides with a well-defined structure are involved, in the other only small molecules. Please explain better this concern and provide a better link between traditional amyloidosis and IEMs.

Response: Thank you for the assessment. We have explained a link between traditional amyloid aggregation and IEMs in revised draft.

Comment 6. The table 2 is difficult to read. Please reorganize the table by summarizing the results and not providing full sentences. Please pay attention to typography. In the X-ray part don't forget to provide the PDB code.

Response: Thank you for pointing this out. As per the reviewer’s opinion, we have reorganized the table. It is difficult to include the PDB codes for the crystal structures of amyloidogenic proteins since we could not find them in literature. Moreover, due to polymorphic nature of amyloid one PDB code in general for a particular amyloid cannot be ascertained. However, several research papers have reported the x-ray diffraction data of amyloidogenic peptide or protein excluding the PDB codes previously.

Comment 7. In the conclusion section, the authors should be careful and not give inappropriate considerations. A common etiology or a common origin between amyloidosis and IEMs are not the correct conclusions. This might be true for the hereditary forms but for the others? The sentence lines 564-571 is too long and difficult to understand.

Response: Thank you for the valuable suggestion. We have revisited and reorganized the conclusion section.

Minor suggestions by the reviewer:

- line 72: is it not 220 nm?

Response:  Thank you for this excellent suggestion. We are sorry for not mentioning the accurate and appropriate CD data of amyloid. We have changed it to 220 nm.

- paragraph 3: don't forget to explain the difference between Abeta 1-40 and Abeta 1-42 by citing the work of Xiao et al. where a three-strand beta sheet structure of Abeta 1-42 is provided as possible explanation of the fast aggregation propensity.

Author response: Thank you for this suggestion. We have explained the difference between Aβ42 and Aβ40 in revised draft. It will be found from lines 135-140.

- line 173: a full stop after the alpha letter

- line 231: space

Response: Thank you and kindly accept our apologies for these mistakes. We have revised the manuscript.

- line 356: is it abeta 1-4à or 1-42? In the brakets are the numbers of residues?

Response: Thank you for pointing this out. In the brackets 11-25 represent the number of residues.

It is not Aβ-42. It is the fragments of amyloid-β peptide and these fragments serve as the valuable systems of the core amyloid structure.

- line 386: two times the ref 126

- lines 451-469: organisation of the paragraph.

- Table 2 line 473: it should be table 1 and it should be present a link with the text. It is not commented.

Response: Thank you and kindly accept our apologies for these mistakes. We have revised the manuscript.

Reviewer 2 Report

The review is devoted to the description of amyloid fold as a universal marker of pathogenicity in multiple amyloid-related diseases. The review lists several pathologies and provides detailed information on amyloid structures related to these pathologies. Also, the ability of single amino acids to form amyloids is acknowledged by a separate section.

Some improvements must be done.

The review does not mention about the existence of functional amyloids (for example yeast adhesins). However, functional amyloids should be mentioned in the review, at least in order to demonstrate that not all amyloids are detrimental. It could be done in discussion section. Also, the discussion section would benefit from mentioning of approaches to prevent of reduce disease-related amyloid formation. The authors mentioned inhibitors of amyloid formation. They should mention the idea of utilization of specific anti-amyloid or anti-prion antibodies to prevent amyloid formation. Also, the idea of using of anti-prion systems to prevent amyloid formation could be mentioned as well. 

Page 1. Lines 40-42. “Research on studying the progression of plethora of prion -diseases suggests formation of infectious prion protein (PrP) with a tendency to cross seed aggregation in other proteins. ” 

This sentence needs in editing. If the authors meant to say that infectious PrP protein induces aggregation or even prion behavior of other proteins then references to original research articles should be provided demonstrating examples of such activities.

Section 5 should include a clear definition of a prion. What is the difference between a prion and amyloid? It is not clear after reading section 5.

Figure 4. Depositions of amyloids should be shown using arrows. In figure 4 legend, the left and the right part seem to be misplaced. Also, the photo in the left part of the figure 4 can not be found in the reference 121.

Page 11. Line 377. An excessive dot should be removed from the sentence.

Line 389. An excessive reference 126 should be removed as well.

The reference 124 is omitted in the list of references used.

Figure 5 should be clarified. The reference 131 mention in Figure 5 legend is not related to p53 protein studies and should be replaced by relevant reference. Also, the references to original research articles mentioning the connection of aggregation of p53 with all the diseases mentioned in Figure 5 should be provided.

Table 2 could be improved by the addition of the addition of 3rd column with the concentration range at which amino acids mentioned are capable of amyloid formation. Also it would be nice to know whether the amyloid formation by each amino acid can happen in aqueous solutions.

Comment to Section 8. The cases (publications) when anybody reported the formation of amyloid deposits based on single amino acids in case of amyloid diseases in living organisms should be referenced and discussed.

Page 16. Line 558. The table on this page has wrong number.

The review can be accepted after major revision.

The manuscript should be checked for typos.

Author Response

Response to Reviewer 2 comments to the authors:

Reviewer 2

The review is devoted to the description of amyloid fold as a universal marker of pathogen city in multiple amyloid-related diseases. The review lists several pathologies and provides detailed information on amyloid structures related to these pathologies. Also, the ability of single amino acids to form amyloids is acknowledged by a separate section.

Some improvements must be done.

Comment 1. The review does not mention about the existence of functional amyloids (for example yeast adhesins). However, functional amyloids should be mentioned in the review, at least in order to demonstrate that not all amyloids are detrimental. It could be done in discussion section. Also, the discussion section would benefit from mentioning of approaches to prevent or

 reduce disease-related amyloid formation. The authors mentioned inhibitors of amyloid formation. They should mention the idea of utilization of specific anti-amyloid or anti-prion antibodies to prevent amyloid formation. Also, the idea of using of anti-prion systems to prevent amyloid formation could be mentioned as well. 

Response: Thank you for this suggestion. We have added one new section for functional amyloids to address the reviewer’s concern. Section 9 represents the functional amyloids.

We have discussed the therapeutic approaches to prevent amyloid formation in conclusion section.

We have incorporated the importance of antibodies against amyloidogenic diseases in section 10.

Comment 2. Page 1. Lines 40-42. “Research on studying the progression of plethora of prion -diseases suggests formation of infectious prion protein (PrP) with a tendency to cross seed aggregation in other proteins.” This sentence needs in editing. If the authors meant to say that infectious PrP protein induces aggregation or even prion behavior of other proteins then references to original research articles should be provided demonstrating examples of such activities.

Section 5 should include a clear definition of a prion. What is the difference between a prion and amyloid? It is not clear after reading section 5.

Response: Thank you for this valuable comment. We have added two references (9,10) to discuss the cross-seeding behavior of prion protein.

We have provided a definition of prion and explained in detail about the difference in between prion and amyloid in the revised draft.

Comment 3. Figure 4. Depositions of amyloids should be shown using arrows. In figure 4 legend, the left and the right part seem to be misplaced. Also, the photo in the left part of the figure 4 cannot be found in the reference 121.

Response: Thank you for this suggestion. We have modified all the figures.

Comment 4. Page 11. Line 377. An excessive dot should be removed from the sentence.

An excessive dot is removed.

Comment 5. Line 389. An excessive reference 126 should be removed as well.

The excessive reference is removed in the list of references in the revised draft

Comment 6. Figure 5 should be clarified. The reference 131 mention in Figure 5 legend is not related to p53 protein studies and should be replaced by relevant reference. Also, the references to original research articles mentioning the connection of aggregation of p53 with all the diseases mentioned in Figure 5 should be provided.

Response: Thank you for this suggestion. We have modified all figures in revised text.

Comment 7. Table 2 could be improved by the addition of the addition of 3rd column with the concentration range at which amino acids mentioned are capable of amyloid formation. Also, it would be nice to know whether the amyloid formation by each amino acid can happen in aqueous solutions.

Response: Thank you very much for the comment. The research pertaining to amyloid formation is in progress. So far, amyloid-like structures are reported for phenylalanine, tyrosine, tryptophan, cysteine, and methionine. The concentrations of amyloid like structure formation as observed by microscopy in these papers are in the range 1 to 10mM which is significantly higher than the physiological concentration which ranges in micromoles. Most of the studies conducted are in vitro and done within a specific time period of incubation. Hence, it is very difficult to compare it with physiological concentrations as it occurs after long time and the chemical environment present is completely different.  Hence there is no literature which suggests exact concentration at which single amino acids will form amyloid like structures inside the body and in vivo studies coupled with clinical trials on animal models will be required for it. A testing on human model is still a far-reaching vision. Hence, we have not included in table the concentration at which single amino acids from amyloid since it is at hypothesis stage and single amino acid-based amyloids are yet to be identified inside the body.

Comment 8. Comment to Section 8. The cases (publications) when anybody reported the formation of amyloid deposits based on single amino acids in case of amyloid diseases in living organisms should be referenced and discussed.

Response: Thank you for your assessment! There is no publication wherein amyloid like deposits for single amino acids were observed in living organism till now. The studies need to be conducted in this direction in future.

Comment 8. Page 16. Line 558. The table on this page has wrong number.

Response: We have changed the table number.

The review can be accepted after major revision.

Comments on the Quality of English Language

The manuscript should be checked for typos.

Response: The manuscript has been checked for typos.

Round 2

Reviewer 1 Report

The authors provided convincing answers and corrections and they improved their review. I just recommend to insert in Ref 191 a recent review on cross-interactions (Molecules 2020, 25(10), 2439; https://doi.org/10.3390/molecules25102439).